# Combination of Serum and Plasma Biomarkers Could Improve Prediction Performance for Alzheimer’s Disease

**DOI:** 10.3390/genes13101738

**Published:** 2022-09-27

**Authors:** Fan Zhang, Melissa Petersen, Leigh Johnson, James Hall, Sid E. O’Bryant

**Affiliations:** 1Institute for Translational Research, University of North Texas Health Science Center, Fort Worth, TX 76107, USA; 2Department of Family Medicine, University of North Texas Health Science Center, Fort Worth, TX 76107, USA; 3Department of Pharmacology and Neuroscience, University of North Texas Health Science Center, Fort Worth, TX 76107, USA

**Keywords:** Alzheimer’s disease, blood biomarkers, support vector machine, machine learning, feature selection

## Abstract

Alzheimer’s disease (AD) can be predicted either by serum or plasma biomarkers, and a combination may increase predictive power, but due to the high complexity of machine learning, it may also incur overfitting problems. In this paper, we investigated whether combining serum and plasma biomarkers with feature selection could improve prediction performance for AD. 150 D patients and 150 normal controls (NCs) were enrolled for a serum test, and 100 patients and 100 NCs were enrolled for the plasma test. Among these, 79 ADs and 65 NCs had serum and plasma samples in common. A 10 times repeated 5-fold cross-validation model and a feature selection method were used to overcome the overfitting problem when serum and plasma biomarkers were combined. First, we tested to see if simply adding serum and plasma biomarkers improved prediction performance but also caused overfitting. Then we employed a feature selection algorithm we developed to overcome the overfitting problem. Lastly, we tested the prediction performance in a 10 times repeated 5-fold cross validation model for training and testing sets. We found that the combined biomarkers improved AD prediction but also caused overfitting. A further feature selection based on the combination of serum and plasma biomarkers solved the problem and produced an even higher prediction performance than either serum or plasma biomarkers on their own. The combined feature-selected serum–plasma biomarkers may have critical implications for understanding the pathophysiology of AD and for developing preventative treatments.

## 1. Introduction

Alzheimer’s Disease (AD) is the most common cause of dementia accounting for 60–80% of cases. Currently, more than 6 million Americans have AD and by 2050 an estimated 13 million Americans will suffer from it [1]. Currently, there is no cure, but the development of biologically based screening could facilitate early diagnosis and enhance early intervention. Body fluid biomarkers and positron emission tomography (PET) imaging of AD might closely reflect synaptic dysfunction in the brain [2]. For example, PET amyloid pathology (Aβ PET)) and the three body fluid biomarkers––Aβ, total tau (t-tau), and phosphorylated tau (p-tau)—have risen in prominence [3]. However, cerebrospinal fluid (CSF) and PET examinations are far from standard tests. High cost, invasiveness or insufficient accessibility, might limit their applications [4]. Niklas et al. binarized the AT(N) markers (A: CSF Aβ42 and amyloid-PET; T: CSF phosphorylated tau and tau PET; and N: hippocampal volume, temporal cortical thickness, and CSF neurofilament light (NfL)) and found that using different AT(N) variants might cause important prognostic information to be lost because they were not interchangeable, and that optimal variants differ by clinical stage [5]. Blood-based biomarkers represent a significant alternative to a primary care-based screening algorithm for AD, given the high prevalence of the disease [3,6,7,8,9,10,11].

Doecke et al. identified a panel of plasma biomarkers that distinguish ADs from normal controls (NCs) with performance of 85% for sensitivity and specificity and 93% for area under the receiver operating characteristic curve (AUC) [12]. They validated their panel using the AD neuroimaging initiative (ADNI) cohort with 112 Ads and 58 NCs and attained 80% accuracy for sensitivity and specificity and 85% for AUC. Ray et al. identified 18 signaling proteins in blood plasma that could discriminate the ADs from NCs with 90% positive predictive value (PPV) and 88% negative predictive value (NPV) [13]. O’Bryant et al. identified 23 serum protein biomarkers and used an optimal random forest risk score to achieve comparable performance (AUC = 91%; sensitivity = 80%; and specificity = 91%) [10].

Most AD biomarker identification is conducted using only one type of blood fraction [3,6,7,8,9,10,11,12,13]. Although both serum and plasma have been used in developing blood-based biomarker profiles, serum has been used more frequently and has demonstrated higher sensitivity in AD detection [6,7,8,9,10,11] than plasma. Combined assays from each of the blood fractions has not been used in blood-based biomarker profiling. Our hypothesis is that combining biomarkers in serum- and plasma-based assays may help to identify a candidate set of protein biomarkers with higher confidence.

Our analyses will use machine learning to assess the efficacy of combining serum and plasma biomarkers. However, due to the high complexity of machine learning, simply combining them can cause overfitting, where the model does much better on the training set than on the testing set [14]. Adding more features doesn’t necessarily increase model performance especially in a blind testing set because too many input features will end up memorizing noise instead of finding the signal. Therefore, we employed a feature selection method to reduce the number of features and select the proper combination of serum and plasma biomarkers to improve the model’s performance [15].

In this paper, we first tested to see if simply adding the biomarkers can increase prediction performance but cause overfitting. Then we performed a feature selection for the combined biomarkers. After feature selection, we tested a combination of serum and plasma biomarkers to see if it yielded highest performance. We performed this test 10 times in a repeated 5-fold cross validation model for training and testing sets.

## 2. Materials and Methods

### 2.1. Serum and Plasma Data Collection

Blood serum and plasma samples were analyzed and evaluated clinically with 150 AD and 150 NC cases for serum and 100 AD and 100 NC cases for plasma. Briefly, each participant at one of the five participating members of the Texas Alzheimer’s Research and Care Consortium (TARCC) sites underwent an annual standardized assessment, that included a medical evaluation, neuropsychological testing and a blood draw. The five consortium members were Baylor College of Medicine, Texas Tech University Health Sciences Center, University of North Texas Health Science Center, University of Texas Southwestern Medical Center, and University of Texas Health Science Center at San Antonio. The AD diagnosis was based on National Institute of Neurological and Communicative Disorders and Stroke and the Alzheimer’s Disease and Related Disorders Association (NINCDS-ADRDA) criteria [16,17], which has shown good reliability and validity [18]. Criteria for AD was based on impairment in neuropsychological assessment in eight specified cognitive domains as well as a decline in functional abilities. Normal controls were those whose psychometric testing performance fell within normal limits defined as expected performance given age and education level but did not meet the NINCDS-ADRDA criteria of cognitive impairment.

### 2.2. Assay

Samples were prepared for proteomic analysis with the Hamilton Robotics StarPlus system. Serum and plasma samples were assayed through a multiplex biomarker assay platform using electrochemiluminescence (ECL). A Meso Scale Discovery Electrochemiluminescence QuickPlex 120 imager used electrodes to introduce a charge (“electro” component) into each well of the MSD ELISA plate. The chemiluminescence component was the emission of light during a chemical reaction. The MSD ECL ELISA used the electrical charge to alter the state of Ruthenium (II) tris-bipyridine-(4-methylsulfone) [Ru(bpy)3] conjugated detection SULFO-TAG antibodies, making them chemically active. Ru(bpy)3-based tag underwent a rapid redox reaction that emitted light in the presence of tripropylamine (TPA) a co-reactant for light generation when voltage is applied creating a electro-chemical reaction producing a luminescent (light emitting) response.

A total of 500 µL of serum and plasma was obtained from each TARCC sample and used to assay the following markers: fatty acid binding protein 3 (FABP3), β 2 microglobulin (B2M), pancreatic polypeptide (PPY), C-reactive protein (CRP), Thrombopoietin (TPO), α 2 macroglobulin (A2M), eotaxin 3, tumor necrosis factor α (TNFα), soluble tumor necrosis factor receptor-1 (sTNFR1), tenascin C, interleukin (IL)-5, IL-6, IL-7, IL-10, IL-18, I-309, Factor VII (Factor 7), soluble intercellular adhesion molecule-1 (sICAM-1), circulating vascular cell adhesion molecule-1 (sVCAM-1), thymus and activation regulated chemokine (TARC), and serum amyloid A (SAA). The proteins were selected based on previous work validating their utility in spanning platforms, tissues and species [8,11].

### 2.3. 10 Times Repeated 5-Fold Cross-Validation

The k-fold cross-validation is a standard method for model performance estimation in machine learning. A common value for k is 5. Here, the dataset was split into 5 folds. In each iteration, one of the folds is used to test the model and the rest are used to train the model. This process is repeated until each fold has been used as the testing set. A single run of the 5-fold cross-validation procedure may result in a noisy estimate of model performance. Different splits of data may result in very different results. Therefore, repeated k-fold cross-validation provides a way to improve the estimated performance of a machine learning model. We set the repeat times at 10. It simply repeated the 5-fold cross-validation procedure 10 times and averaged the performances across all folds from all runs. Six performance metrics were measured in expectation that they would give a more accurate estimate of the true unknown underlying mean performance of the model on the dataset.

### 2.4. Performance Measurement

The following six metrics were involved in our performance evaluation: sensitivity, specificity, precision, accuracy, negative predictive value, and area under the curve.
(1)$$Sensitivity=TP/\left(TP+FN\right)$$
(2)$$Specificity=TN/\left(TN+FP\right)$$
(3)$$Precision=TP/\left(TP+FP\right)$$
(4)$$Accuracy=\frac{TP+TN}{TP+TN+FP+FN}$$
(5)$$NPV=TN/\left(TN+FN\right)$$

### 2.5. Feature Selection

A recursive feature elimination algorithm based on a support vector machine (SVM) model and a cross-validation that we developed [15,19] was used for selecting serum and plasma biomarkers. Briefly, the feature selection procedure involved four steps: training an SVM on the training set, calculating ranking criteria based on the SVM performance, eliminating features with the smallest ranking criteria, and repeating the process [15]. This feature selection method used a leave-one-out cross-validation, where the fold number equaled the sample size.

## 3. Results

Of the 150 AD serum, 150 normal control serum samples and the 100 AD plasma and 100 normal control plasma samples, only 79 ADs and 65 NCs both had serum and plasma samples in common. We conducted a two-step statistical power analysis to determine the sample size for a significance level and power to protect against Type I errors. First, we calculated the power or sample size with the power package in R(v4.2.1) for two sample Student’s *t*-tests. Then we determined the machine-learning power, given the sample size and the significance level with the pROC package in R(v4.2.1). For the sample sizes of 150 and 100, the power was 0.994 and 0.956, respectively, for the serum and plasma datasets. For the unequally merged sample size of 79 ADs and 65 NCs, the power was 0.836. Using sample size and power computation for the ROC curves, the final sample sizes presented above would provide an observed power of 1 for AD vs. NC analysis with α = 0.05 for detecting AUC > 0.95. In each sample, 21 protein biomarkers were measured for expression values. Table 1 shows the demographic-characteristics of the samples for both serum and plasma. The NC group was significantly younger than the AD group (*p* < 0.05). However, there was no significant difference in sex or education between them (*p* < 0.05).

### 3.1. Recursive Feature Elimination

The core of the feature elimination algorithm was to repeat removing the features that had the lowest ranking criteria until the error rate did not decrease. The least number of features with the lowest error rate was found at 10. After feature elimination, 4 serum (Serum_IL6, Serum_IL7, Serum_sTNFR1, and Serum_sVCAM1) and 6 plasma biomarkers (Plasma_TPO, Plasma_Eotaxin3, Plasma_SAA, Plasma_IL6, Plasma_sTNFR1, Plasma_sICAM1) were selected (Figure 1 and Table 2). The Spearman’s correlation of the selected 10 serum and plasma biomarkers showed that the recursive feature elimination properly detected and eliminated features that were highly correlated (Figure 2a,b), for example: Plasma_I309 and Serum_I309 (corr = 0.8959), Plasma_CRP and Serum_CRP (corr = 0.8987), Plasma_IL10 and Serum_IL10 (corr = 0.9007), Plasma_SAA and Serum_SAA (corr = 0.9091), and Plasma_FABP3 and Serum_FABP3 (corr = 0.9252).

The correlation coefficients for serum and plasma sTNFR1 and IL6 were 0.763290732 and 0.721979744, respectively. We defined a strong, moderate, or weak correlation if the absolute value of correlation coefficient was between 0.8 and 1, 0.6 and 0.8, and below 0.6, respectively. The coefficients for serum and plasma sTNFR1 and IL6 would be on the boundary line if a correlation coefficient were between 0.7 or 0.75 and 1 was considered to be a strong correlation. We added the following leave-one-out experiments for four markers (serum_sTNFR1, plasma_sTNFR1, serum_IL6, and plasma_IL6) to evaluate the performance of removing each of them. The 10 times repeated 5-fold cross-validation average performance for a testing set of 15 ADs and 13 NCs were all precision/PPV 82.35, accuracy 85.71, sensitivity 93.33, specificity 76.92, and NPV 90.91 except that the four AUCs were 94.73%, 94.95%, 92.87%, and 93.92%, respectively. This result showed that removing any one of them would not improve prediction performance for the testing set (*p* = 0.038). This result validated the feasibility of the recursive feature elimination method.

We further compared recursive feature elimination with principal component analysis (PCA) to reduce the dimensionality of the dataset for machine learning. Setting the number of principal components in PCA as 10, the 10 times repeated 5-fold cross-validation average performance for the testing set of 15 ADs and 13 NCs became Precision/PPV 75.00, accuracy 75.00, sensitivity 80.00, specificity 69.23, NPV 75.00, and AUC 81.62%, which didn’t outperform the recursive feature elimination method (*p* = 0.001). This might be because the PCA was unsupervised learning and didn’t consider the target variable when reducing the dimension.

### 3.2. Serum Only vs. Plasma_Only vs. Serum + Plasma vs. Serum + Plasma + Feature Elimination

We compared prediction performance in the 10 times repeated 5-fold cross validation model for 21 serum biomarkers only (serum_only), 21 plasma biomarkers only (plasma_only), and the combined biomarkers (Serum + Plasma). Both kinds of biomarkers were selected by the feature elimination algorithm (Serum + Plasma + Feature elimination). The model was used to increase the number of estimates and improve the accuracy of the prediction model by avoiding overfitting. The confusion matrix was averaged from the cross-validation and rounded to calculate the performance.

The SVM importance score for the selected 4 serum and 6 plasma biomarkers in the 10 times repeated 5-fold cross-validation model is shown in Figure 3. The SVM importance scores were used to determine the most contributory features for SVM classifiers. They were calculated using the importance method in rminer package R(v4.2.1). For example, Serum_IL6 ranked #1 and Plasma_sICAM1 ranked #10, which was consistent with results from the single-variable analysis (Serum_IL6 *p* = 0.001 and Plasma_sICAM1 *p* = 0.545) in Table 2. Plasma_SAA in Table 2 had a higher *p*-value of 0.673 but ranked #4 in the SVM importance score, which further verified the interesting finding that insignificant variables could be selected in the final serum and plasma mixture as we show in the Section 4.

Table 3 and Table 4 show that serum biomarkers yielded better performance than plasma biomarkers in both the training and the testing sets of the 10 times repeated 5-fold cross-validation model (*p* = 3.7 × 10^−4^, *p* = 1.21 × 10^−4^, for the training and testing sets respectively). When combining serum and plasma biomarkers, the prediction performance improved only in the training set (*p* = 0.03 for the training and *p* = 0.36 for the testing sets, respectively) because increasing the number of features more likely led to overfitting. We performed a recursive feature elimination algorithm to avoid this problem when integrating serum and plasma biomarkers. After feature elimination, the selected 4 serum and 6 plasma biomarkers yielded the highest performance (*p* = 0.042 and *p* = 0.019, for the training and testing sets, respectively: Table 3 and Table 4). The final ROC curve for the selected serum and plasma biomarker combination is shown in Figure 4.

### 3.3. Serum + Plasma vs. Serum Only + FeatureElimination vs. Plasma Only + Feature Elimination

We further compared the prediction performance of the combined serum and plasma biomarkers with the feature selection on the biomarkers individually and found that the combination yielded a higher prediction performance. This could have been caused by serum and plasma biomarker integration as predictors in experiments to overcome the heterogeneity of AD.

### 3.4. Age as Covariate

We also experimented by adding age as a covariate to the selected 10 serum and plasma biomarkers. The average performance for the 10 times repeated 5-fold cross-validation model with age as a covariate improved in the training set with Precision/PPV 98.46, accuracy 99.14, sensitivity 100.00, specificity 98.08, NPV 100.00, and AUC 99.96% (*p* = 0.02). However, the average performance for the testing set remained the same as for the selected 10 serum and plasma biomarkers.

## 4. Discussion

### 4.1. Challenges of Integrating the Serum and Plasma Data

Preventing overfitting is a big challenge when integrating two types of data to predict AD because of this disease’s nature and heterogeneity, the relationship between the different data (serum and plasma), and their interactions. The challenge demands the development and application of new analysis strategies to integrate them.

In this paper, we first used the 10 times repeated 5-fold cross-validation model to detect overfitting. For example, we found that the training set performance improved after the serum and plasma biomarkers were mixed, but the performance for testing set did not improve. Then we applied the feature selection algorithm to overcome the fitting problem by reducing the number of mixed serum and plasma biomarkers from 42 to 10. The prediction performance of the testing set improved from a specificity of 69.23 to 76.92% while keeping the sensitivity of 100% unchanged.

When we integrated serum and plasma data, we sometimes not only faced the problem of overfitting but other issues such as data transformation or normalization. If the two types of data did not come from standardized data, the transformation (weighting) of data might be required. Fortunately, both the serum and plasma data came from same protocol and data transformation in our database, so we only needed to deal with overfitting.

### 4.2. Insignificant Variables Selected in the Final Serum and Plasma Mixture

After feature selection, we identified 4 serum (Serum_IL6, Serum_IL7, Serum_sTNFR1, and Serum_sVCAM1) and 6 plasma biomarkers (Plasma_TPO, Plasma_Eotaxin3, Plasma_SAA, Plasma_IL6, Plasma_sTNFR1, Plasma_sICAM1). Previous studies showed that they are of clinical significance when tested individually. For example, Interleukin-6 (IL-6) acts as both a pro-inflammatory cytokine and an anti-inflammatory myokine. It has important clinical roles in both innate and adaptive immunity [20] and has been proven to play a role in the pathogenesis of AD. Studies found that the elevation of peripheral IL-6 was associated with increased risk of developing Alzheimer’s disease [21,22,23].

Interleukin-7 (IL-7) stimulates the proliferation of pre-B and pro-B cells without affecting their differentiation. In human peripheral monocytes, IL-7 induces the synthesis of some inflammatory mediators such as IL-1 and IL-6. Rakic et al. performed a post-mortem human study to determine whether systemic infection modified the neuropathology and found that in AD it was associated with decreased IL-7 [24].

Soluble Tumour Necrosis Factor Receptor 1 (sTNFR1) clinically identifies diabetic patients at high risk of end stage renal disease (ESRD) up to 10 years in advance. Diniz et al. found that a high serum sTNFR1 level was associated with a higher risk of progression from mild cognitive impairment (MCI) to AD [25]. Hu et al. showed that higher levels of proteins related to sTNFR1 were associated with a reduced risk of conversion to dementia and that these sTNFR1-related inflammatory proteins provided prognostic information independent of established Alzheimer’s markers [26].

Soluble ICAM-1 (sICAM1) and VCAM-1 (sVCAM1) have clinical significance as markers of endothelial activation [27]. Endothelial dysfunction can lead to a worsening of cognitive performance in AD. Huang et al. measured plasma levels of VCAM-1, ICAM-1 and found them to be significantly higher in AD patients. They concluded that endothelial activation, especially VCAM-1, reflected macro- and micro-structural changes and poor short term memory and visuospatial function [28].

Eotaxin3 are small proteins included in the group of chemokines. They have clinical use in inflammatory diseases such as allergic diseases and cancer [29]. They also show an emerging role in neurodegenerative disease [30]. The ADNI study (http://adni.loni.ucla.edu (accessed on 22 September 2022)) found that increased levels of eotaxin 3 were associated with AD [31].

An Australian research group assessed mean biomarker levels for sVCAM1, sICAM1, TPO eotaxin3 and SAA. over 36 months and found there to be an association between biomarker levels and clinical classification [32].

In our dataset, six of the 10 biomarkers were not significantly different statistically between AD and NC. For example, Plasma_SAA, Plasma_IL6, Serum_sTNFR1, Serum_sVCAM1, Plasma_sTNFR1, and Plasma_sICAM1 had *p* > 0.05 (Table 2). This showed that serum or plasma variables that are not significantly different can be selected by a machine-learning method to improve prediction performance. When combined, the 4 serum and 6 plasma biomarkers demonstrated their prediction power for clinical application.

Furthermore, we dug a little deeper into the false intuition that accuracy is always directly proportional to the number or importance of features. First, machine learning is as much an experimental science as a theoretical one. In general, there is no universal rule stating that the accuracy of a machine learner is directly proportional to the number or importance of features used to train it. Take an example of linear regression: Predicting the price of a house based solely on square footage is not going to be a very accurate measure of the real value of the house. Augmenting it with the number of bedrooms may give a much more realistic price estimate than augmenting it with the lot area although the lot area as a sole variable has s higher importance score than the number of bedrooms for the price of a house.

Second, AD is not a single homogeneous disease but multiple disease states, each arising from a distinct molecular mechanism and having a distinct clinical progression path that makes the disease difficult to detect and predict in the early stages.

Third, a panel of proteins rather than single protein candidate may have greater implications and collectively they may predict the disease and its pathology better [33,34]. In the panel, some insignificant genes might be important in some significant pathways towards and away from AD [35]. Feature selection based on performance measurement rather than only on importance scores is a feasible way to extract AD-related biomarker networks for better prediction.

### 4.3. Limitations

#### 4.3.1. Small Sample Size

There are two possible limitations that could be addressed in future research. First, the sample size in the testing set was only 28, which is relatively small. A small sample size would cause problems, for example, by causing machine learning to lose power and accuracy. It would also made detection of overfitting difficult. For example, the accuracy from Serum_only to Serum + Plasma + Feature Elimination increased only 3.58% from 85.71 to 89.29%. We had to look at other metrics such as the NPV which increased 10% from 90.91 to 100% to determine the importance of overfitting reduction if we followed the acceptable NPV FDA requirement: NPV greater than 97%. For our future research, we planned to collect about 500 samples from serum and plasma in the Health and Aging Brain Study—Health Disparities (HABS-HD) project [36,37,38,39].

#### 4.3.2. Overfitting Measurement

The second limitation was that there is currently a lack of a decisive threshold for determining model overfitting, but it can be actually seen as a relative measurement. In this study, we used the change rate to measure performance improvement to catch the overfitting. For example, Table 3 and Table 4 showed that combining serum and plasma (serum + plasma) improved the training set performance. For example, accuracy improved 0.9% from 96.55% in Serum_only to 97.41% in Serum + Plasma) but not for the testing set (accuracy unchanged from 85.71% from Serum_only to Serum + Plasma). Overfitting could be the culprit because some highly correlated variables were combined for the Serum + Plasma model (for example, Plasma_I309 and Serum_I309 (corr = 0.8959), Plasma_CRP and Serum_CRP (corr = 0.8987), Plasma_IL10 and Serum_IL10 (corr = 0.9007), Plasma_SAA and Serum_SAA (corr = 0.9091), and Plasma_FABP3 and Serum_FABP3 (corr = 0.9252)).

After we performed the recursive feature elimination, overfitting became less. For example, Table 3 and Table 4 showed that serum and plasma after elimination (Serum + Plasma + Feature Elimination) improved performance not only for the training set (for example, accuracy in Serum + Plasma improved 0.9% from 97.41 to 98.28% in Serum + Plasma + Feature Elimination) but also for the testing set (e.g., accuracy improved 4.2% from 85.71% in Serum + Plasma to 89.29% in Serum + Plasma + Feature Elimination).

To further test whether combining serum and plasma caused overfitting problem and whether recursive feature elimination reduced overfitting, we measured the Mean Squared Error (MSE) for each model and its standard deviation: 0.140 ± 1.64, 0.353 ± 2.13, 0.149 ± 1.62, and 0.105 ± 1.39 for Serum_only, Plasma_only, Serum + Plasma, and Serum + Plasma + Feature Elimination, respectively. The recursive feature elimination for Serum + Plasma alleviated both MSE and variance. This was another sign that overfitting problem in mixed serum and plasma was alleviated via the recursive feature elimination method.

In future research, we might look into advanced analytical methods for measuring the overfitting problem, such as a relative change rate, which can be defined as the change rate relative to the size of sample.

## 5. Conclusions

In this paper, we tested the hypothesis that combining serum and plasma biomarkers with feature selection would improve prediction performance for AD. First, we determined if simply adding serum and plasma biomarkers could not only improve prediction performance but also cause overfitting. We then employed a feature selection algorithm we developed to overcome the problem while maintaining the higher prediction performance of serum and plasma biomarkers mixed together. Lastly, we tested the prediction performance in a 10 times repeated 5-fold cross validation model for both the training and testing sets. We found that a combination of selected serum and plasma biomarkers after feature selection yielded a higher prediction performance than either a combination of only serum biomarkers or a combination of only plasma markers. The combined serum–plasma biomarker approach is useful for overcoming the many challenges associated with data-driven heterogeneity analyses of AD.

## Figures and Tables

**Figure 1 genes-13-01738-f001:**
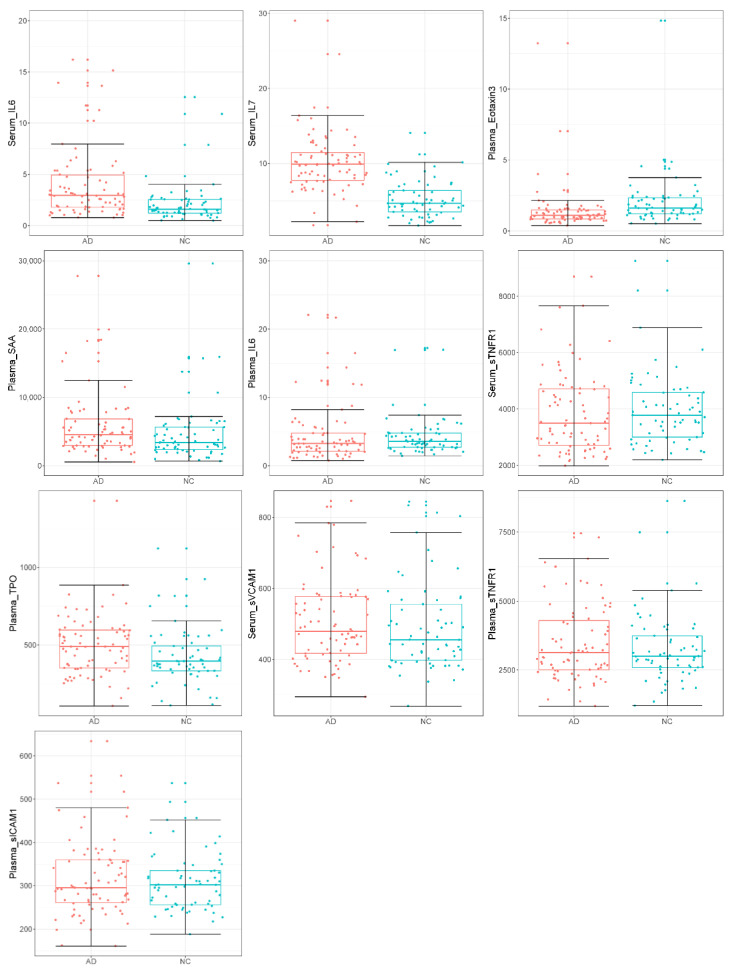
Boxplots for the selected 4 serum and 6 plasma biomarkers between 79 ADs and 65 NCs (red for ADs; cyan for NCs).

**Figure 2 genes-13-01738-f002:**
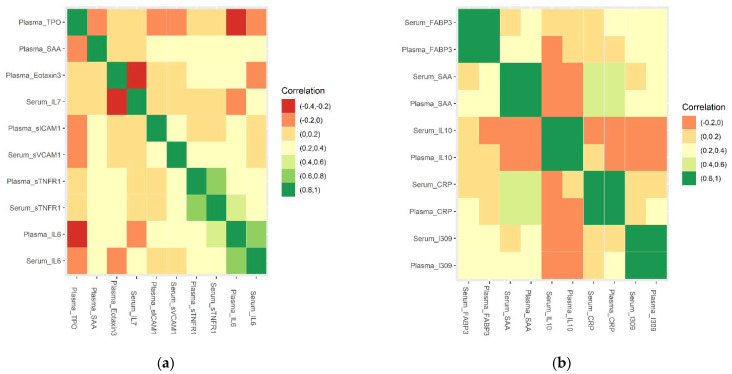
Correlation of selected 10 serum and 10 plasma biomarker combinations (**a**) selected 10 serum and plasma biomarkers (**b**) unselected 5 pairs of serum and plasma biomarkers that were highly correlated.

**Figure 3 genes-13-01738-f003:**
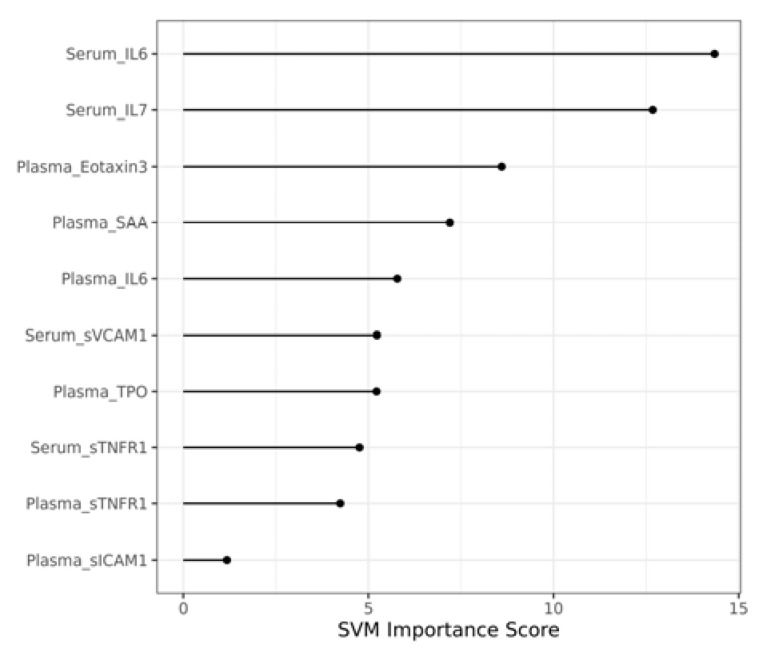
Importance score for selected 10 serum and plasma biomarker combination in the 10 times repeated 5-fold cross-validation model.

**Figure 4 genes-13-01738-f004:**
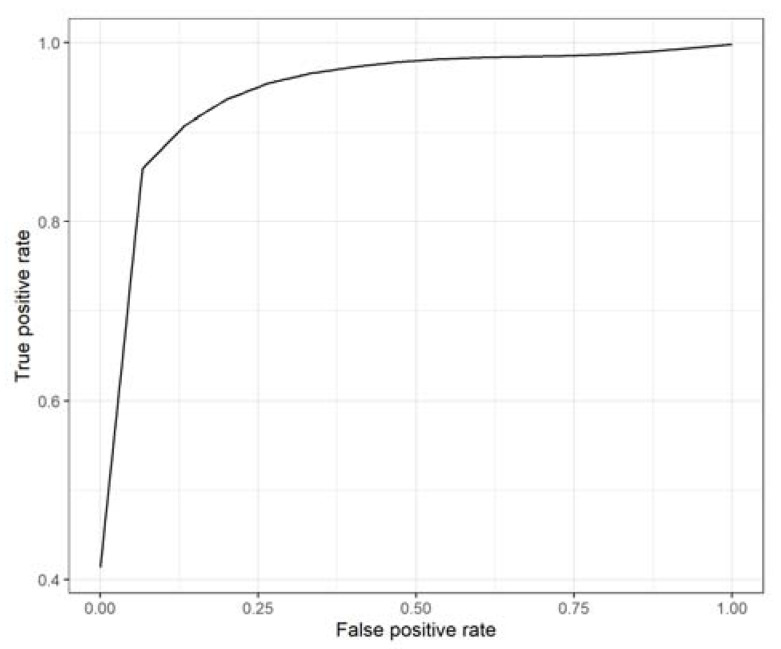
ROC curve for selected 10 serum and plasma biomarker combinations in the 10 times repeated 5-fold cross-validation model.

**Table 1 genes-13-01738-t001:** Demographic characteristics of the cohort.

	AD Mean (SD)	Normal Control Mean (SD)	*p*-Value
N	79	65	
Age	76.14 (8.79)	71.57 (8.91)	0.002
Education	14.61 (3.07)	15.72 (2.63)	0.020
Sex (% M)	30.4	32.3	0.806

**Table 2 genes-13-01738-t002:** Protein changes in the selected serum and plasma biomarker panel.

Protein	GeneID	NC	AD	Direction	*p*-Value
Serum_IL7	3574	5.26	10.15	up	7.23 × 10^−15^
Serum_IL6	3569	2.19	4.40	up	0.001
Plasma_TPO	7173	424.31	495.23	up	0.026
Plasma_Eotaxin3	10,344	2.06	1.43	down	0.036
Plasma_sTNFR1	7132	3261.06	3468.76	up	0.352
Plasma_IL6	3569	4.43	5.12	up	0.413
Serum_sVCAM1	7412	491.83	508.53	up	0.426
Serum_sTNFR1	7132	3988.99	3848.72	down	0.542
Plasma_sICAM1	3383	310.35	318.40	up	0.545
Plasma_SAA	6287	8345.07	10,110.06	up	0.673

**Table 3 genes-13-01738-t003:** Average performance for training set of 64 ADs and 52 NCs in the 10 times repeated 5-fold cross-validation model.

	Serum		Plasma		Serum + Plasma		Serum + Plasma + Feature Elimination	
Predicted	AD	NC	AD	NC	AD	NC	AD	NC
AD mean ± sd	64 63.88 ± 0.34	4 4.46 ± 1.22	61 60.66 ± 1.49	8 8.05 ± 1.97	64 64.00 ± 0.00	3 2.57 ± 1.34	64 64.00 ± 0.03	2 1.70 ± 1.01
NC mean ± sd	0 0.12 ± 0.34	48 47.54 ± 1.22	3 3.34 ± 1.49	44 43.95 ± 1.97	0 0.00 ± 0.00	49 49.43 ± 1.34	0 0.00 ± 0.03	50 50.30 ± 1.01
Precision/PPV	94.12%		88.41%		95.52%		96.97%	
Accuracy	96.55%		90.52%		97.41%		98.28%	
Sensitivity	100.00%		95.31%		100.00%		100.00%	
Specificity	92.31%		84.62%		94.23%		96.15%	
NPV	100.00%		93.62%		100.00%		100.00%	
AUC	99.55%		97.25%		99.98%		99.16%	

**Table 4 genes-13-01738-t004:** Average performance for the testing set of 15 ADs and 13 NCs.

	Serum		Plasma		Serum + Plasma		Serum + Plasma + Feature Elimination	
Predicted	AD	NC	AD	NC	AD	NC	AD	NC
AD mean ± sd	14 14.37 ± 0.86	3 3.30 ± 1.49	12 11.74 ± 1.57	7 6.64 ± 1.71	15 14.54 ± 0.64	4 3.71 ± 1.54	15 14.57 ± 0.71	3 2.52 ± 1.27
NC mean ± sd	1 0.63 ± 0.86	10 9.70 ± 1.49	3 3.26 ± 1.57	6 6.36 ± 1.71	0 0.46 ± 0.64	9 9.29 ± 1.54	0 0.43 ± 0.71	10 10.48 ± 1.27
Precision/PPV	82.35%		63.16%		78.95%		83.33%	
Accuracy	85.71%		64.29%		85.71%		89.29%	
Sensitivity	93.33%		80.00%		100.00%		100.00%	
Specificity	76.92%		46.15%		69.23%		76.92%	
NPV	90.91%		66.67%		100.00%		100.00%	
AUC	92.78%		70.91%		93.96%		95.99%	

## Data Availability

The data was obtained from TARCC and are available at https://www.txalzresearch.org/research/data-requests/ (accessed on 22 September 2022) with the permission of TARCC.

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
