# Peer review of "Combination of Serum and Plasma Biomarkers Could Improve Prediction Performance for Alzheimer’s Disease"

_genes, 2022, doi:10.3390/genes13101738_

Round 1

Reviewer 1 Report

In this paper “Combination of serum and plasma biomarkers could improve prediction performance for Alzheimer's disease” the authors tested both serum and plasma biomarkers together to improve prediction performance and overcome overfitting problems. They employed a combined feature-selected serum-plasma biomarker approach for understanding the pathophysiology of AD and for developing its preventative treatments. In my opinion, this paper requires major attention to the result section as figure number and reference are presented as errors. Corrections are required. Along with this problem, there are few suggestions that may be included to improve this manuscript:

1)      In the result section, the authors emphasized on the overcoming overfitting problem. More clear explanation of it will be helpful.

2)      Among all the biomarkers “fatty acid binding protein 3 (FABP3), beta 2 microglobulin (B2M), pancreatic polypeptide (PPY), C-reactive protein (CRP), Thrombopoietin (TPO), alpha 2 macroglobulin (A2M), eotaxin 3, tumor necrosis factor-alpha (TNFα), soluble tumor necrosis factor receptor-1 (sTNFR1), tenascin C, interleukin (IL)-5, IL-6, IL-7, IL-10, IL-18, I-309, Factor VII (Factor 7), soluble intercellular adhesion molecule-1 (sICAM-1), circulating vascular cell adhesion molecule-1 (sVCAM-1), thymus and activation regulated chemokine (TARC), and serum amyloid A (SAA)”, authors mentioned the reason for choosing the 10 biomarkers. There are several literatures mentioning the importance of the rest of the biomarkers.  A justification for not selecting other biomarkers is required.

3)      The Spearman’s Correlation of the selected 10 serum and plasma biomarkers showed that the recursive feature elimination properly detected and eliminated the features that were highly correlated. Was the same high correlation coefficient observed for the control?  A more detailed explanation will be helpful to conceptualize it.

4)      Font is different in the Figure captions of figures 2,3,4.

Reviewer 2 Report

The authors in this work reported the results obtained from the combination of serum and plasma biomarkers that could improve the prediction performance in Alzheimer’s disease.

These results and the topic are interesting and, although I do not have so much expertise in machine learning/algorithms approach, the methodologies are sound.

However, I have some major points to address:

1. the authors should explain better the statistical analyses introducing a dedicated paragraph.

2. The authors should explain better how they calculated the power analysis: they wrote that 100 subjects are enough to have a power of 95%, however, their final sample size is < 100.

3. To be clearer, the authors could divide the Results section in subparagraphs, which could underline the major aims of the work.

4. As a “error reference source” is often present in the text (lines 166, 179, 182, 188, 196, 197, 204, 205 217, 219), it is difficult to follow the results obtained.

5. Thus, Figure 1, Figure 3, Table 4 and Table 5 were not explained in the text.

Table 4 and Table 5 are not clear: did the authors perform the analyses on 64 ADs and 52 NCs/15 ADs and 13NCs respectively? Why? What’s the meaning of Figure 3? What’s the SVM score?

6. The whole paragraph from “Secondly,…..” to “…for 10 times repeated 5-fold cross-validation model” (page 5 lines 199-220) is not so clear to me.

7. In the Discussion section, at page 10 lines 306-307 I do not understand where the authors have found these results.

8. The authors could add a Limitations section.

Minor points:

1. In the Tables 3, 4, check for PPP.

2. Check for acronymous: for instance, interleukin-6 (page 11 line 319), Alzheimer’s disease (page 11 line 322 …), MCI (page 11 line 334), Tumour Necrosis Factor receptor 1 (page 11 line 331)

3. Ckeck for English errors/typos: for instance, “eurogenerative disease (line 347), “Hu et al showed higher levels of proteins related to sTNFR1 were associated ….; page 11 lines 334-335)

Reviewer 3 Report

The manuscript "Combination of serum and plasma biomarkers could improve prediction performance for Alzheimer's disease" by Fan Zhang  et al, presented a study to test if combinination of seurm and plasma biomarkers will cause overfitting problem and if feature selection can overcome the overfitting problem in Alzheimer's disease prediction. However, the result didn't show preformance improvement by combining serum and plamsa biomarkers for prediction in test dataset, and using feature selection actually slightly improved the performance by correct only 1 wrong predicted sample. My suggestion is major revision.

1) Authors didn't check their manuscript carefully. For example, "investigation, X.X.;" in Author Contributionn. All the table/figure references in manuscript are broken.All of them were displayed as "Error! Reference source not found". It's hard to find correct linked table/figure.

2) The training only model should be removed since cross-validation should be the mininum requirement, especially when there were training set result of the 10 times repeated 5-fold cross-validation model in manuscript.

3) In figure2, two figures used different colors for same correlation bins, which might misleading readers.Also, since the paired markers in figure 2b were displayed together, the paired markers in figure 2a should be displayed together too.

4) In table 2, the markers were not ordered. Suggest to order them by p-value.

5) Based on table 4 and table 5, combining Serum and Plasma assigned one more sample in training set and same number of sample in testing set correctly, comparing to Serum only. It is hard to call that combining Serum and Plasma caused overfitting problem.

5) In table 5, the only difference between Serum+Plasma+FeatureElimination and Serum only is one wrong assigned AD sample.Also, in the final 10 markers, there are still two identical genes existed in both serum and plasma (sTNFR1 and IL6) which are highly correlated.I will suggest authors to have a try on dimension reduction instead of feature elimination to improve the prediction performance.

6) Authors should release their data and their code for validation by reviewers and readers.Although authors indicated that "The datasets for this study can be found in the datasets at https://apps.unthsc.edu/itr/.", I cannot find it.

Round 2

Reviewer 1 Report

No Comment. 

Author Response

thank you.

Reviewer 2 Report

The authors answered my comments correctly

Author Response

thank you.

Reviewer 3 Report

The authors solved most of my concerns. The only request that authors should solve is, the data and code should be publicly available through github or other services, instead of send request to authors. Once this request is solved, the paper should be accepted.
